# Copolymerization of Norbornene and Styrene with Anilinonaphthoquinone-Ligated Nickel Complexes

**DOI:** 10.3390/polym11071100

**Published:** 2019-06-28

**Authors:** Samiul Islam Chowdhury, Ryo Tanaka, Yuushou Nakayama, Takeshi Shiono

**Affiliations:** Graduate School of Engineering, Hiroshima University, Hiroshima 739-8527, Japan

**Keywords:** nickel catalyst, copolymerization, norbornene, styrene

## Abstract

Poly(norbornene-*co*-styrene)s were synthesized by the use of anilinonaphthoquinone-ligated nickel complexes [Ni(C_10_H_5_O_2_NAr)(Ph)(PPh_3_): **1a**, Ar = C_6_H_3_-2,6-*^i^*Pr; **1b**, Ar = C_6_H_2_-2,4,6-Me; **1c**, Ar = C_6_H_5_] activated with modified methylaluminoxane (MMAO) or B(C_6_F_5_)_3_ in toluene. The effects of the cocatalysts were more significant than those of the nickel complexes, and MMAO gave higher activity than B(C_6_F_5_)_3_. The structural characterizations of the products indicated the formation of statistical norbornene copolymers. An increase of the styrene ratio in feed led to an increase in the incorporated styrene (S) content of the resulting copolymer. The molecular weight of the copolymer decreased with increasing the S ratio in feed at 70 °C. The copolymerization activity, using MMAO as a cocatalyst, decreased with lowering of the temperature from 70 to 0 °C, accompanied by an increase in the molecular weight of the copolymer. The S incorporation up to 59% with *M_n_* of 78,000 was achieved by the **1b-**B(C_6_F_5_)_3_ catalytic system. The glass transition temperatures of the norbornene (N)/S copolymers determined by differential scanning calorimetry, decreased from 329 to 128 °C according to the S content.

## 1. Introduction

The polymerization of cycloolefins such as norbornene (N) has been one of the key developments in the area of polymer chemistry because of its vast range of applications [1,2,3]. There are three types of mechanism for polymerization of N, which are ring-opening metathesis polymerization (ROMP) [4,5], cationic or radical polymerization [6,7,8,9], and coordination–insertion polymerization [10,11,12]. Each route of the polymerization leads to the polymers with different structures and properties. Polynorbornenes (PNBs) produced by coordination–insertion polymerization show excellent physical properties, such as good heat and chemical resistance, high decomposition temperature, high optical transparency, and low dielectric constant [13,14], but exhibit some negative properties such as brittleness and glass transition temperature (*T_g_*) close to the decomposition temperature due to the presence of a rigid ring in the polymer chain.

The inferior properties of PNBs can be improved in cyclic olefin copolymers (COCs). The most representative COC is a copolymer of N and ethylene (E) [15]. The properties of COCs can be easily controlled by a kind of comonomer, comonomer content, and sequence distribution. Copolymerization of N with 1-alkene was also reported to modify the physical properties [16,17,18,19,20,21,22,23,24]. The introduction of styrene (S) should decrease the *birefringence* of COCs [25] because PNB and polystyrene possess positive and negative *birefringence,* respectively. Several examples of the copolymerization of N with S have been reported. The first N/S copolymerization was achieved by Ni-based catalysts using methylaluminoxane (MAO) as a cocatalyst [26]. Afterwards, some nickel and copper catalysts were used to synthesize N/S copolymers [27,28,29,30,31,32,33]. The S incorporation was improved by β-diketiminato nickel complexes [34]. The highest S incorporation was 52.4%, with the highest S feed ratio of 83%, but the *M_w_* value was around 10^3^. Recently, bis(β-ketoamino) copper complexes were used for copolymerization of N and S, whereas the results were almost the same as those of β-diketiminato nickel complexes [35]. 

We have also reported N/S copolymerization by an *ansa*-fluorenylamidodimethyltitanium-based catalyst and shown the potentiality of the copolymer as a plastic substrate for flexible display materials [25]. However, the maximum S incorporation in the N/S copolymer with sufficient molecular weight was approximately 5 mol %. We have synthesized N/E/S terpolymer by using the same catalyst and obtained a zero-*birefringence* terpolymer with the S content of 12%, but the introduction of E unit (58 mol %) caused a significant drop of the *T_g_* to 86 °C [36].

Therefore, we are interested in the copolymerization of N with S to obtain high molecular weight copolymers with controlled S content. We have previously reported that anilinonaphthoquinone-ligated nickel complexes activated with B(C_6_F_5_)_3_ as a cocatalyst exhibit high activity for N polymerization to give high molecular weight polymer soluble in cyclohexane [37]. In the present paper, we report the N/S copolymerization using nickel complexes [Ni(C_10_H_5_O_2_NAr)(Ph)(PPh_3_): **1a**, Ar = C_6_H_3_-2,6-*^i^*Pr; **1b**, Ar = C_6_H_2_-2,4,6-Me; **1c,** Ar = C_6_H_5_], where **1c** was newly synthesized in this work, in the presence of modified MAO (MMAO) or B(C_6_F_5_)_3_ (Figure 1).

## 2. Materials and Methods

### 2.1. Materials

All manipulations were carried out under a nitrogen atmosphere using standard Schlenk techniques. All solvents were refluxed and distilled over sodium/benzophenone or calcium hydride. Norbornene was purified by stirring it over calcium hydride at 60 °C for one day, and then distilled. The stock solution of N (5.5 M) was prepared in toluene. Styrene (Wako Chemical Co. Ltd., Odawara, Japan) was dried over CaH_2_, and then freshly distilled under vacuum prior to use. Modified methylaluminoxane (MMAO) solution (6.6 wt % in toluene) and toluene solution of B(C_6_F_5_)_3_ were donated from Tosoh Finechem. Co. (Tokyo, Japan) and used as received. The nickel complexes **1a,1b** were synthesized according to the literature and the references therein [38,39]. The complex **1c** was synthesized using a similar procedure to that for **1a** and **1b** (Appendix A).

### 2.2. Analytical Procedure 

Molecular weights and molecular weight distributions of polymers were determined by gel permeation chromatography (GPC) with a Viscotec HT-350 GPC (Malvern, Great Malvern, UK) with one guard column and two 30 cm columns. This system was equipped with a triple-detection array consisting of a differential refractive index (DRI) detector, a two-angle (7, 90) light scattering (LS) detector, and a four-bridge capillary viscosity detector. Polymer characterization was carried out at 150 °C using *o-*dichlorobenzene as an eluent, and calibrated with polystyrene standards. The ^1^H and ^13^C NMR spectra of polymers were measured at room temperature on a Bruker 500M Hz instrument (Bruker, Rheinstetten, Germany) operated by the pulse Fourier-transform mode. Sample solution of ^13^C NMR was prepared in CDCl_3_ up to 10 wt %, and the pulse angle was 45°, and about 8000–10,000 scans were accumulated in pulse repetition of 5.0 s. The central peak of CDCl_3_ (7.13 ppm for ^1^H and 77.13 ppm for ^13^C NMR) was used as an internal reference. Differential scanning calorimetry (DSC) was performed on a SII EXSTER 600 system (Seiko Instruments Inc., Chiba, Japan) under nitrogen atmosphere. Thermal history difference in the polymers was eliminated by first heating the specimen to 380 °C, cooling from 10 to 20 °C/min, and then recording the second DSC scan at a heating rate of 10 °C/min. 

### 2.3. Copolymerization of N and S 

In a typical procedure, prescribed amounts of N and S in toluene solution were introduced into a 100 mL round-bottomed glass flask. Then, the cocatalyst (0.24 mL 500 µmol of MMAO toluene solution or 1 mL 20 µmol of B(C_6_F_5_)_3_ toluene solution) and 1 mL of the nickel complex (5 µmol) solution in toluene were syringed into the well-stirred monomer solution in this order, and the total solution volume was made up to 25 mL by adding toluene. The copolymerization was conducted under continuous stirring for a required time under a certain temperature as controlled using an external oil or ice bath. The copolymerization was terminated by adding 300 mL of acidic methanol (methanol/concentrated hydrochloric acid, 95: 5 in volume). The resulting precipitated polymer was collected by filtration, adequately washed with methanol, and dried in vacuum at 60 °C for 6 h. 

## 3. Results and Discussion

### 3.1. Homopolymerization of N and S 

Homopolymerizations of N and S were performed using **1-**MMAO at 70 °C in toluene. The results are shown in Table 1. In the N polymerization, complex **1a** displayed the highest activity, and gave the polymer with the highest molecular weight among the complexes used. The opposite trends were observed in S polymerization, but the differences were not significant. 

### 3.2. Copolymerization of N and S 

Copolymerizations of N with S were then conducted under the same conditions by changing S in feed from 10 to 40 mmol, with 40 mmol of N (Scheme 1).

The results are summarized in Table 1. The nickel complexes showed moderate activity for N/S copolymerization. The N/S feed ratio did not significantly affect the catalytic activity, and all the complexes showed lower activity in the copolymerizations than in homopolymerizations. The low activity in the copolymerization could be ascribed to slow cross-propagation because of the steric hindrance between these comonomers. Complex **1a**, which showed the highest activity in N polymerization, also showed the highest activity in N/S copolymerization among the complexes used (Table 1, Run 3). 

The N/S copolymers were not only soluble in chloroform, but also in cyclohexane, similar to the PNB obtained by the same catalytic system. The incorporation of S in the produced copolymers was investigated by ^1^H NMR in CDCl_3_. A typical ^1^H NMR spectrum of the copolymer (Run 15) is illustrated in Figure 2i. No resonances are observed from 5.0 to 6.0 ppm, which is assigned to the vinylene protons of the polymer obtained via ring-opening metathesis polymerization (ROMP) [40].

The signals assignable to the aliphatic protons of the N and S units (H_N_^1–4^ and H_S_^5^) are observed in the range of 0.8–2.4 ppm. The signals attributed to the aromatic protons of the S units (H_S_^1–3^) are observed in the range of 6.5–7.1 ppm. Particularly, the signal of the methine proton of the S unit (H_S_^4^) adjacent to the N unit can be observed at 2.85 ppm. The ^1^H NMR spectra of the polymers obtained under the highest styrene concentration (Table 1, Run 3, 9 and 15) are shown in Appendix A. 

A typical ^13^C NMR spectrum of the copolymer (Run 15) is shown in Figure 2ii. According to the reported assignment of the N/S copolymer, [34] the signals of each chemical shift region were assigned as follows: 145.7 ppm for C^1^, 127.3 ppm for C^2^, 127.9 ppm for C^3^, 125.6 for C^4^, 41.5–44.2, and 40.5 ppm for C^5^ and C^6^ of the S segment, 47.1–52.5 ppm for C^3^, 38–39.7 ppm for C^2^, 34.5–37 ppm for C^4^, and 29.4–31.9 ppm for C^1^ of the N unit. A clear observation of the phenyl-carbons of the S units in the regions of 125–145 ppm also indicates the presence of the S units in the N/S copolymer. These results are similar to those obtained in the previously reported nickel–MAO catalytic system and confirm the random distribution of S unit in the obtained N/S copolymer [34]. 

The incorporation of S increased with increasing S concentration in the monomer feed ratio. The highest S content in the N/S copolymer was achieved to be 36% at 1:1 feed ratio by complex **1c** probably because of less interaction between the ligand substituent and the aromatic ring of S. 

The molecular weight of N/S copolymer was measured by GPC. The increase of S concentration in feed caused the decrease in the molecular weights of the produced polymers accompanied by the increase in the number of polymer chains, which is ascribed to β-hydrogen elimination of the increased styryl propagation end [41]. Unimodal distribution (*M_w_/M_n_* ≈ 2) indicates that the copolymerization should take place at a single active site (Appendix A).

### 3.3. Effect of Polymerization Temperature on N/S Copolymerization:

The influence of reaction temperature on N/S copolymerization was studied using **1-**MMAO at the fixed feeding ratio (N/S = 4:1 in molar ratio), because the low S feed ratio gave high molecular weight N/S copolymer. The results are summarized in Table 2. 

The copolymerization activity decreased with lowering the temperature from 70 to 0 °C. The highest activity around 70 kg polymer/(mol Ni h) for the copolymerization was achieved at 70 °C in each system. On the other hand, the S content and the *M_n_* value monotonously increased with lowering of the polymerization temperature (Appendix A), and the N/S copolymer with 32 mol % of S and 61,000 of *M_n_* was obtained at 0 °C by **1c-**MMAO (Table 2, Run 25, Figure 3i). This *M_n_* value would be the highest among the N/S copolymers reported so far [25,34,35]. The molecular weight distribution became narrow with decreasing the polymerization temperature. A decreased *M_n_* value of the copolymers obtained with an increase in the reaction temperature is ascribed to the chain transfer at high temperature [32].

The *T_g_* values of N/S copolymers were analyzed by DSC. The *T_g_* value declined with an increase of S content as shown in Figure 3ii. The highest value of 329 °C (Appendix A), and the lowest value of 172 °C, were detected for the copolymers with 4 and 32 mol % of S, respectively (Table 2, Run 6 and 25). A previous study has shown that the *T_g_* value of the PNBs obtained with this catalyst was over 400 °C [37]. The *T_g_* value of polystyrene is 100 °C. The single *T_g_* value in the DSC curves reveal that N and S were uniformly distributed in the N/S copolymers obtained by the present catalysts. 

### 3.4. N/S Copolymerization by 1-B(C_6_F_5_)_3_:

We found the considerable effects of temperature on N/S copolymerization using the **1-**MMAO system. We observed significant effects of B(C_6_F_5_)_3_ in E polymerization [38] and N polymerization [37] with **1a**, which were ascribed to the formation of zwitterionic nickel species [39]. 

Thus, the N/S copolymerization (N/S = 4:1 in molar ratio) by the **1-**B(C_6_F_5_)_3_ system, was conducted at different temperatures (30, 50, and 70 °C) to investigate the cocatalyst effect. 

The activity of the B(C_6_F_5_)_3_ system was lower than that of the MMAO system. The incorporation of S increased with lowering the temperature, as was observed in the **1-**MMAO system. The S incorporation was almost four times higher than those of the copolymers produced by the **1-**MMAO system. The highest S incorporation was observed at 30 °C (Appendix A), 51~59% with the *M_n_* values (Appendix A) of 43,000~78,000 (Table 3, Run 28, 31, and 34).

In order to evaluate the effects of the cocatalysts on N/S copolymerization, the monomer reactivity ratios of **1b** were determined at 70 °C (Appendix A). The Fineman–Ross plots of **1b**-MMAO and **1b**-B(C_6_F_5_)_3_ for N/S copolymerization are shown in Figure 4i and 4ii, respectively. The monomer reactivity ratios were determined to be *r*_N_ = 6.34 and *r*_S_ = 0.39 for **1b**-MMAO, and *r*_N_ = 0.93 and *r*_S_ = 0.13 for **1b**-B(C_6_F_5_)_3_, indicating the better copolymerization ability of the B(C_6_F_5_)_3_ cocatalyst. The reactivity ratios of **1c**-MMAO were also determined to evaluate the effects of the complexes (Appendix A). The values *r*_N_ = 2.16 and *r*_S_ = 0.26 indicate the better copolymerization ability of **1c** than **1b**. 

The *T_g_* values determined by DSC are shown in Table 3 and are plotted against S content in Figure 3ii. The plot is in good accordance with that of the MMAO system, but extends to the higher S content as indicated by the monomer reactivity ratios. The lowest *T_g_* value of 128 °C (Appendix A) was obtained by the copolymer with S content of 59 mol % and *M_n_* of 78,000 (Table 3, Run 31). The **1**-B(C_6_F_5_)_3_ system gave the copolymers with high molecular weights and high S contents, compared with the **1**-MMAO system, although the activity was low. 

The lower activity would be due to the deactivation by impurities in the absence of scavenging reagents. Therefore, we conducted N/S copolymerization with **1a**-B(C_6_F_5_)_3_ in the presence of the reaction mixture of *^t^*Bu_3_Al and 2,6-*^t^*Bu_2_-*p*-cresol. The addition of the scavenger increased the activity, keeping the same S content and the *M_n_* value reached 103,000 (Table 3, Run 35). The results indicate the potentiality of **1**-B(C_6_F_5_)_3_ for N/S copolymerization.

### 3.5. Optical Property of the N/S Copolymer 

The light transmittance of the N/S copolymer thin film with the thickness about 100 µm is displayed in Appendix A (Table 3, Run 35). The copolymer showed the transmittance above 85% in the visible light region (300–800 nm). 

## 4. Conclusions

Copolymerizations of N and S were achieved by anilinonaphthoquinone-ligated nickel-complexes using MMAO or B(C_6_F_5_)_3_ as cocatalyst. The **1**-MMAO system showed higher activity than the **1**-B(C_6_F_5_)_3_ system, whereas the latter system produced the copolymers with higher molecular weight and higher styrene incorporation. The molecular weights of N/S copolymers obtained by **1**-B(C_6_F_5_)_3_ were the highest among those of the copolymers reported previously using nickel-, copper-, and titanium-based catalytic systems.

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
