# Peer review of "Copolymerization of Norbornene and Styrene with Anilinonaphthoquinone-Ligated Nickel Complexes"

_polymers, 2019, doi:10.3390/polym11071100_

Round 1
Reviewer 1 Report
Manuscript ID : polymers-531424
Title : Copolymerization of Norbornene and Styrene with Anilinonaphthoquinone-ligated Nickel Complexes
The article describes the application of a series of nickel precursors used in combination with MMAO or B(C6F5)3 as catalysts of copolymerization of norbornene with styrene.
There are a few minor points/comments which can be addressed prior to the acceptance of this manuscript:
1) The use of MMAO, instead of regular MAO, is not reasoned.
2) Also, the nature of the modified MMAO is not specified. This is a Me3Al-depleted version (“dried” MAO), or a BHT treated MAO, or something else ?
3) The decrease of MW of copolymers upon increasing the styrene feed was explained by a more frequent beta-H elimination as chain termination reaction (lines 154-156). At the same time, no unsaturated chain ends from vinylidene or vinylene groups were observed in the 1H and 13H NMR spectra even for low molecular weight polymer sample obtained in run 15 (Table 1 and Figure 2). There is also a remark in lines 135-137 of the given manuscript. This point showing contradiction between the observed results and proposed hypothesis must be clarified.
4) In the experiment conducted with 1a-B(C6F5)3 in the presence of iBu3Al/BHT (run 35, Table 3) the molecular weight of polymer Mn significantly grew up while the polydispersity slightly dropped as compared to the samples obtained in the runs without scavenger. What could be the nature of processes resulting in such changes ?
5) Lines 39 and 41: mentioning "polyimides" is irrelevant.
Reviewer 2 Report
Manuscript Title:
Copolymerization of Norbornene and Styrene with ailinonaphthaquinone-ligated Nickel complexes
Manuscript ID : polymers-531424
Recommendation: Publish in Polymers after minor revisions
Comments:
In this work, the authors introduced the synthesis of poly(norbornene-co-styrene)s using anilinonaphthaquinone-ligated nickel complexes with MMAO and B(C6F5)3. In order to achieve high molecular weight copolymers with controlled styrene content, the authors systematically investigated the effect of N/S monomer ratio, temperature, and co-catalysts on the reactivity of polymerization and the molecular weight of yielded copolymers. I believe this paper would benefit to the future study of the synthesis and application of well-controlled poly(norbornene-co-styrene)s.
This manuscript is well-organized, and well-written, and thereby is appropriate for a broad readership. I believe this paper could be accepted by Polymers. However, it would be great if the authors would consider the following minor concerns.
1. In Line 59, the authors talked about that they have synthesized the “N/E/S” copolymer; also in Line 61, they mentioned the effect of E unit on the Tg. But they didn’t clarify what is the structure/the full name of “E”. It would be great to add the full name of “E” in that paragraph.
2. Why the Styrene-alone homopolymers have high reactivity than N/S copolymers?
In Table1, the authors showed the trend that the activity decreased with the increased ratio of N/S, and concluded in Line 132 that “the activity of N polymerization should predominate the coplymerization activity”. However, it was also observed that the S-alone polymerizations also have high activities (Entry 4, 10, 16 in Table 1), which were in general better than any copolymerizations, and even better than N-alone polymerizations for complex 1b and 1c.
Thus, the conclusion the author drew that “the activity of N polymerization should predominate the coplymerization activity” could be arguable. It would be great if the author can add some comments on that.
Author Response
We appreciate the reviewer's comments. Our answers are as follows.
The abbreviation of “E” is defined in line 44.
The text has been revised as follows.
Original: “Complex 1a showed the highest activity among the complexes used (Table 1, Run 3), suggesting that the activity of N polymerization should predominate the copolymerization activity.”
Revision: “, and all the complexes showed the lower activity in the copolymerizations than in the homopolymerizations. The low activity in the copolymerization could be ascribed to slow cross-propagation because of the steric hindrance between these comonomers. Complex 1a which showed the highest activity in N polymerization also showed the highest activity in N/S copolymerization among the complexes used (Table 1, Run 3).”
Reviewer 3 Report
The authors report new results on norbornene-styrene copolymerization, producing copolymers with high styrene incorporation and high molar masses.
In my opinion the paper can be accepted for publication, but some English corrections are required.
Moreover, some formatting is necessary, e.g. indentatin of paragraphs and some misprints need to be corrected, e.g. lines 208 and 211by 1( bold), line 209 norbornene.
Author Response
We appreciate the careful review.
According to the reviewer’s comments, the text has been revised.